# Development and validation of an artificial intelligence model based on liver CSE-MRI fat maps for predicting dyslipidemia

Bo Jiang[1], Weijun Situ[1], Zhichao Feng[2], Jianmin Yuan[2], Yina Wang[3], Xiaofan Chen[1], Xiong Wu[1], Kai Deng[1], Haitao Yang[1], Xiao Xiao[3], Xi Guo[1], Junjiao Hu[1,4]*

1 Department of Radiology, The Second Xiangya Hospital of Central South University, Changsha, Hunan, China, 2 Central Research Institute, United Imaging Healthcare, Shanghai, China, 3 Department of Geriatric Medicine, The Second Xiangya Hospital of Central South University, Changsha, Hunan, China, 4 The Key Laboratory of Biomedical Information Engineering of Ministry of Education, Institute of Health and Rehabilitation Science, School of Life Science and Technology, Xi'an Jiaotong University, Xi'an, Shanxi, China

* hujujiao91@csu.edu.cn

## Abstract

This study aimed to develop and validate an artificial intelligence (AI) model for the non-invasive early detection of dyslipidemia using liver chemical shift-encoded MRI (CSE-MRI) fat maps. An automated AI pipeline was developed to predict abnormalities in four lipid indicators: triglyceride, total cholesterol, low-density lipoprotein cholesterol, and high-density lipoprotein cholesterol. The study utilized 1,757 liver CSE-MRI fat images from 89 patients who underwent MRI scans and contemporaneous blood lipid testing. Transfer learning was applied using several pre-trained networks, including ResNet18, MobileNet, DenseNet, AlexNet, and SqueezeNet. Model performance was evaluated via 8-fold cross-validation, with the optimal model further assessed on a held-out test set using confusion matrices and derived metrics. Significant performance differences were observed among models. The optimal model, based on ResNet18, demonstrated high accuracy in the internal validation set. On the independent test set, this model achieved accuracies of 0.853 for triglyceride, 0.833 for total cholesterol, 0.937 for low-density lipoprotein cholesterol, and 0.936 for high-density lipoprotein cholesterol, with corresponding F1-Scores of 0.885, 0.571, 0.886, and 0.897. The AI model based on liver CSE-MRI fat maps shows high accuracy and generalization in predicting abnormalities for three key lipid indices, validating its potential as an early warning tool for dyslipidemia. Expanding the training dataset could further enhance performance for all lipid indices.

**Data availability statement:** All data underlying the findings described in this manuscript are fully and publicly available without restriction. The dataset has been deposited on figshare, a public data repository, and can be accessed via the following Digital Object Identifier (DOI) and URL: DOI: 10.6084/m9.figshare.30796934 URL: https://doi.org/10.6084/m9.figshare.30796934 This statement confirms that the data supporting the study are openly accessible, in accordance with PLOS Digital Health's data sharing policy.

**Funding:** This work was supported by the Natural Science Foundation of Hunan Province (Grant Numbers: 2025JJ60834, 2022JJ70139), the Clinical Medical Technology Innovation and Guidance Project of Hunan Province (2021SK53510), and the Health Research Project of Hunan Provincial Health Commission (W20243040).

**Competing interests:** The authors have declared that no competing interests exist.

## Author summary

Dyslipidemia, a major risk factor for chronic diseases, is traditionally diagnosed via invasive blood tests. We explored a non-invasive alternative by developing an artificial intelligence (AI) model that predicts blood lipid abnormalities using routine liver MRI scans. Our model analyzes chemical shift-encoded MRI (CSE-MRI) fat maps—images that quantify liver fat content—to forecast levels of triglycerides, cholesterol, and other key lipids. We trained and compared multiple deep learning models, finding that a model based on ResNet18 performed best. It demonstrated high accuracy, particularly in predicting abnormal levels of triglycerides and certain cholesterol types. This approach can provide an "opportunistic screening" tool; when patients undergo abdominal MRI for other reasons, their existing scan data could be simultaneously analyzed to assess dyslipidemia risk, adding value without extra cost or scan time. This work pioneers a new, non-invasive method for early dyslipidemia warning using clinical imaging data.

## Introduction

Dyslipidemia usually refers to elevated levels of cholesterol and/or triglycerides in the serum. In fact, dyslipidemia also generally refers to various dyslipidemia including low HDL-C (high-density lipoprotein cholesterol) syndrome.Abnormal blood lipid metabolism is closely related to the occurrence and development of cardiovascular disease, type 2 diabetes, fatty liver disease, metabolic syndrome, mental disease, gonadal function disease, pregnancy-related disorders and other chronic diseases [1–6].Dyslipidemia has become one of the main reasons for the increased burden of chronic diseases [7]. As a result, blood lipid management faces major challenges in the context of global public health [8].

The routine lipid metabolism markers include serum triglyceride (TG) [9], total cholesterol (TC) [8, 10], low density lipoprotein cholesterol (LDL-C) and high density lipoprotein cholesterol (HDL-C) [10], which are critical for assessing cardiovascular and metabolic disease risk. TG provides energy for cellular metabolism and elevated level of TG is linked to cardiovascular diseases [10, 9]. TC reflects total blood cholesterol, while LDL-C transports cholesterol to tissues, with high level of LDL-C contributing to atherosclerosis and coronary heart disease. HDL-C[9], on the other hand, helps transport cholesterol to the liver, playing a protective role against atherosclerosis. Decreased HDL-C is associated with increased cardiovascular risk and is often seen in conditions like diabetes and liver diseases [1, 2].

At present, the main method of monitoring blood lipids is to determine various blood lipid indexes by drawing blood [9]. With the increasing demand for improved healthcare,people hope to conduct lipid monitoring in more diversified ways.Using non-invasive imaging technology to perform morphological examinations and predict blood lipids can reduce the risks associated with invasive procedures and improve the patient experience. Currently, studies associated with evaluation of blood lipids

by imaging methods are mainly focused on the following two aspects. On the one hand, artificial intelligence (AI) is used to calculate height, weight, age, gene, sleep time and other factors to predict blood lipid levels, however, it cannot be used as a basis for clinical diagnosis [11–13]. On the other hand, the content of liver fat is quantitatively evaluated by magnetic resonance fat quantitative sequence [14] or key section images of ultrasound [15, 16] and combined with AI for prediction. The quantitative magnetic resonance fat sequence includes proton density fat fraction (PDFF) obtained by multi-echo chemical shift coding MRI (CSE-MRI) water fat separation technique and MR spectroscopy (MRS). Studies have shown that the two methods are highly correlated and consistent [17], and both can evaluate the degree of fatty liver disease. However, direct prediction of specific blood lipid indexes by magnetic resonance fat quantitative sequence has not been reported. Therefore, it is worth exploring whether data mining of magnetic resonance fat quantitative sequences can directly provide early warning of blood lipid levels.

With the progress and development of AI technology, AI-aided technology is applied to a variety of medical problems [18–20].This method has been successfully applied to various aspects of intelligent medicine [21]. Intelligent evaluation of dyslipidemia based on unenhanced MRI images facilitates early screening, diagnosis, and dynamic monitoring of lipid metabolism-related diseases, thereby offering a new and valuable contribution to imaging-derived biomarkers of cardio-vascular and metabolic disease risk. Therefore, in this study, we attempt to use liver CSE-MRI fat images to train a binary AI model using various pre-training networks. This model is used to predict whether common blood lipid indexes (TG, TC, LDL-C, HDL-C) are abnormal, thereby expanding the functionality of CSE-MRI images and providing a new method for predicting various blood lipid indexes.

## Methods

### Data source

The liver CSE-MRI fat images used in this study were derived from a study on the relationship between obstructive sleep apnea hypopnea syndrome and liver aging. The research program was approved by the Research Ethics Committee of The ********. Date of approval: November 07, 2022, batch number: (2022) Ethical Review (Research 806). This study follows the principles of the Helsinki Declaration. All patients underwent liver CSE-MRI sequence scans, and fasting blood samples were taken on the same day to determine blood lipid levels, including TG, TC, LDL-C and HDL-C. A total of 89 patients who met the requirements were enrolled, regardless of gender and age. Custom screenshot software Flash-capture1.0.25 was used to capture the CSE-MRI fat images of each patient containing the liver in batches. This facilitated the subsequent application of the model. The screenshot areas do not contain any patient privacy information, and the image resolution was 1328×889. Finally, 1757 CSE-MRI fat images at different liver levels (shown in Fig 1) are obtained, each corresponding to a measured value of TG, TC, LDL-C and HDL-C.

### CSE-MRI scanning

A 3.0-T magnetic resonance system (uMR790, Shanghai United Imaging Healthcare Co., Ltd.) equipped with 12-channel dedicated body coils and 30-channel spinal coils was used. All subjects fasted for 4–6 hours before undergoing CSE-MRI scans and were trained to exhale and hold their breath for more than 20 seconds. The slice thickness is 6mm. Approx-imately 20 images can be obtained based on the patient's body type.The subjects were examined in supine position, and coronal and transverse T2-weighted images were obtained for positioning. During a single breath-holding period of 18 seconds, a six-echo GRE_FACT sequence was used to scan the whole liver. After scanning, water, fat, T2* and R2* images were automatically generated, as well as in-phase and out-of-phase images synthesized from water and fat images. The imaging parameters [22] were as follows: number of echoes: 6, TR: 10.5 ms; 6 TE times: 1.64, 3.1, 4.56, 6.02, 7.48, 8.94 ms; field of view: 400 *300 *192 mm $^3$; flip angle: 3 °; voxel size: 2.38*1.67*8.0mm$^3$. The generated fat map is the target image used for training the prediction model in this study.

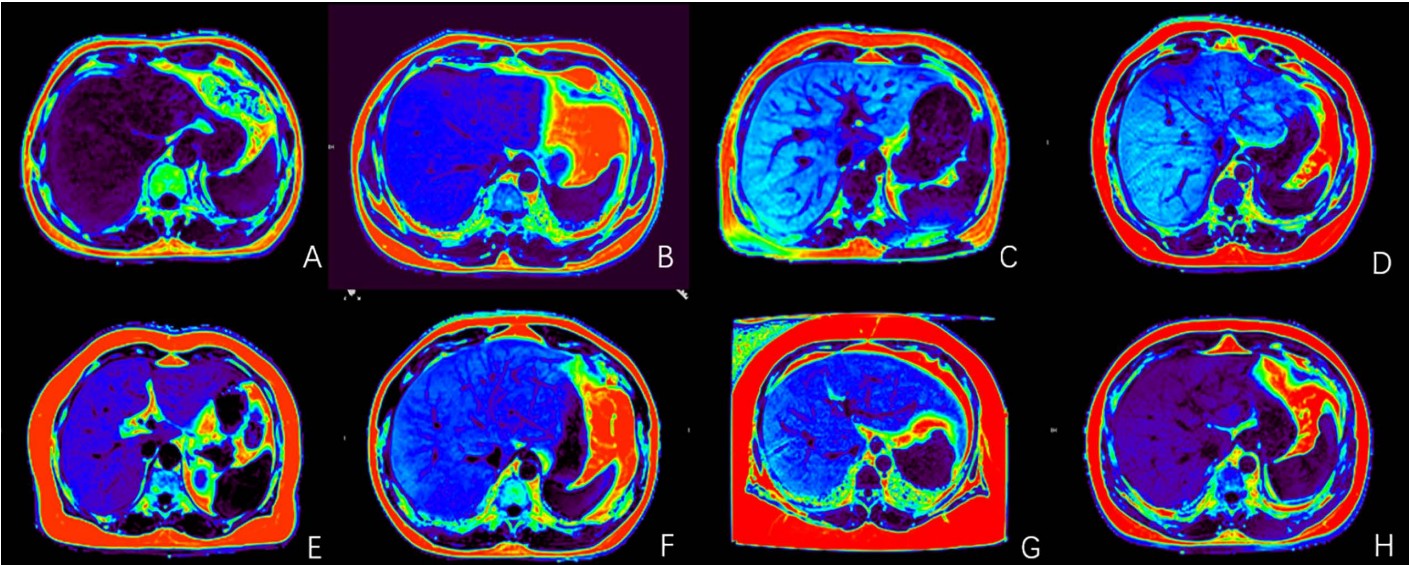

**Fig 1. CSE-MRI fat map of liver corresponding to different blood lipid indexes.** A: CSE-MRI fat map of liver with normal TG. B: CSE-MRI fat map of liver with normal TC. C: CSE-MRI fat map of liver with normal LDL-C. D:CSE-MRI fat map of liver with normal HDL-C. E: CSE-MRI fat map of liver with elevated TG level. F:CSE-MRI fat map of liver with elevated TC level. G: CSE-MRI fat map of liver with elevated LDL-C level. H:CSE-MRI fat map of liver with reduced HDL-C level.

## Image classification and preprocessing

The picture classification is based on whether the serum TG, TC, LDL-C is too high and whether the HDL-C is too low. The critical value refers to the screening standard for ASCVD low-risk population proposed by the latest Chinese guide [10] (all samples are from China).When the TG value is greater than or equal to 1.7mmol/L, it is judged to be too high, and the picture label is set to 1, otherwise it is 0. When the measured value of TC is greater than or equal to 6.2mmol/L, it is judged to be too high, the picture label is set to 1, and vice versa; when the measured value of LDL-C is greater than or equal to 4.1mmol/L, it is judged to be too high, and the picture label is set to 1, and vice versa; when the measured value of HDL-C is lower than 1.00mmol/L, it is judged to be too low, and the picture label is set to 1, and vice versa; among them, there are 628 images with normal TG and 1129 images with too high TG. There were 1511 normal images in TC, 246 images with too high TC, 1491 images with normal LDL-C, 266 images with too high LDL-C, 1021 images with normal HDL-C and 736 images with low HDL-C. In the image preprocessing step, Using the relevant dependency support libraries, write the corresponding code to read the images and their classification labels, use the Scikit-Image (Skimage) library to adjust the size of all images to 221×221 (the original picture size is close to an integer multiple of that size, which helps to retain the original image information in the scaling process), and use the NumericalPython (NumPy) library to generate a dataset in.npy format. The dataset was divided into 1560 TG training sets and 197 TG test sets. There were 1595 samples in the TC training set and 162 samples in the TC test set, 1615 samples in the LDL-C training set and 142 samples in the LDL-C test set, 1616 samples in the HDL-C training set and 141 samples in the HDL-C test set. The images in the test set were separated based on patient boundaries before organizing the image counts, ensuring that all images and related images of patients in the test set do not appear in the training set. This is also the reason for the inconsistency in the number of test set images for different indicators.The specific picture classification and data distribution are shown in the figure (Fig 2).

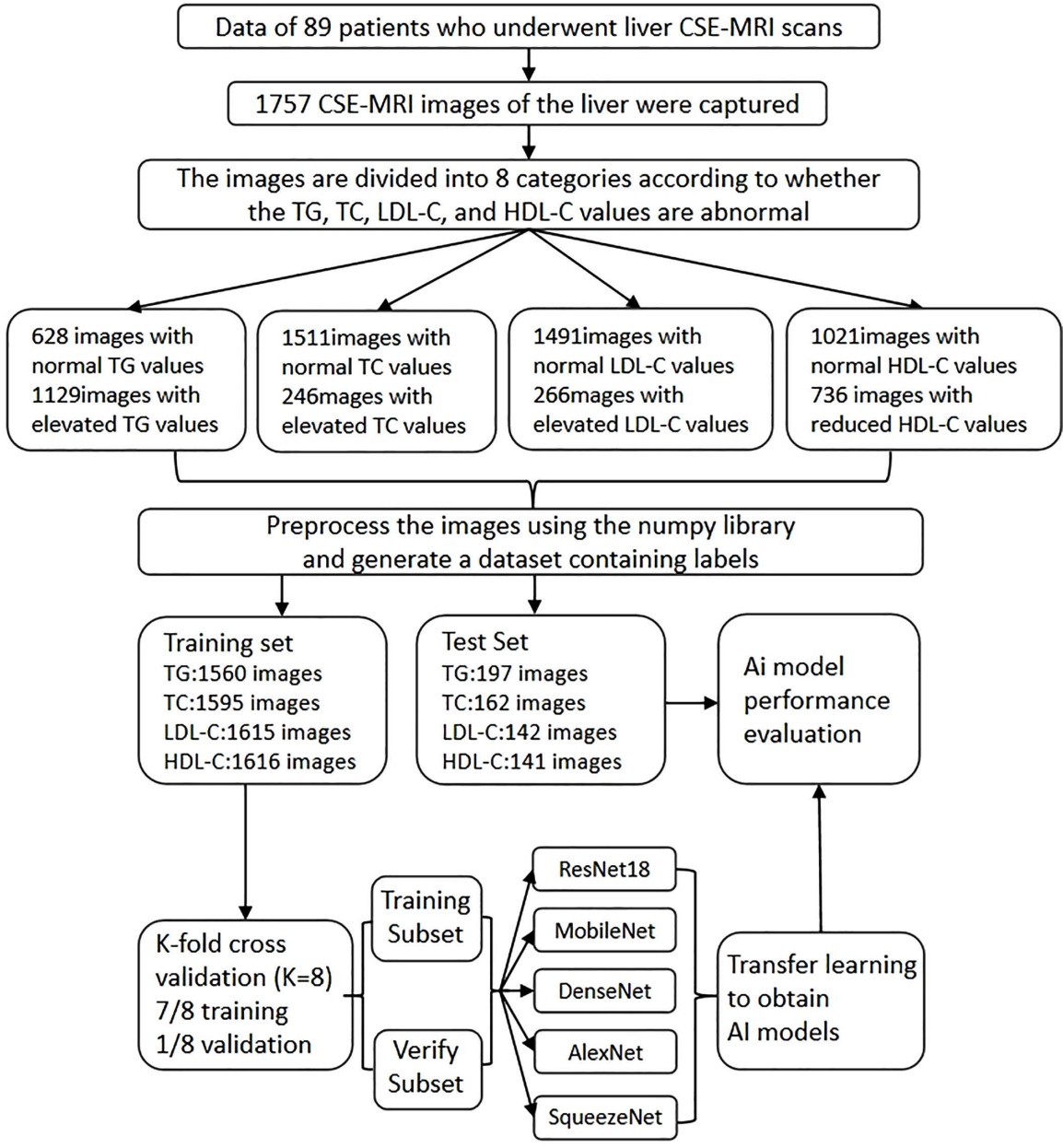

**Fig 2. Flowchart of image classification and dataset allocation.**

## Model training

The open source programming language Python is used to develop the AI algorithm. Using the PyTorch library (version 2.1.0), we performed transfer learning on pre-trained networks including ResNet18, MobileNet, DenseNet, AlexNet, SqueezeNet and other lightweight medical image classification networks are migrated and trained [23–25]. The specific training process is shown in Fig 3. The training process uses the computer equipment of the post-processing center of the Department of Radiology of the second Xiangya Hospital of Central South University, and the software and code were run on a graphics computing server equipped with a V100 16G passive GPU. The model was trained using the Adam

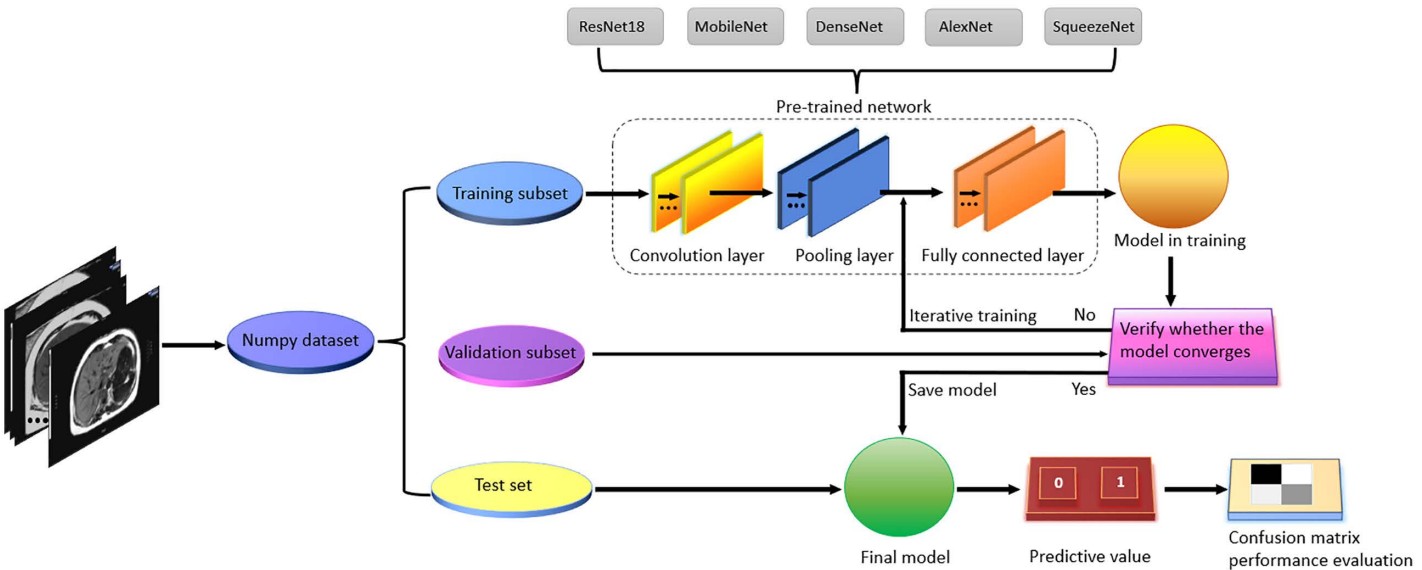

**Fig 3. Flow chart of mobility learning training Ai model and model performance evaluation for different pre-training networks.**

adaptive learning rate optimizer [26], the initial learning rate is 0.0001. L2 regularization function is used to prevent model overfitting, and the parameter value is 0.01 L2 regularization will not make the model sparse while preventing excessive parameters [27]. Too much batch size per input leads to insufficient learning details, while too low batch size may cause the model to overfit [28]. After many tuning, it is found that the accuracy of the model is improved and the loss function is further reduced when batch size is 6. Using 40 iterations epoch, the models generated after each iterative learning are saved respectively. Finally, the model with the lowest loss function and the highest accuracy on the internal verification data set is selected as the final model.

## Model evaluation and statistical analysis

In order to evaluate the performance of each training model as objectively as possible, the data is allocated by k-fold cross-validation (CV) [29], and the accuracy and loss function of the validation set are printed in real time during model training. Finally, the accuracy of classification prediction and the average value of loss function and standard deviation are calculated as comparison indexes between different models. The performance of each model was evaluated in the test set images, and the subroutines of accuracy_score, f1_score, precision_score and recall_ score in scikit-learn1.2.1 library were used to calculate the average value of all models produced by 8 times k-fold CV to compare the differences of different pre-training networks. The scipy.stats library's friedmanchisquare function was used to analyze the statistical differences of the optimal models generated by five pre-training networks on the test set, and based on statistical differences, the scikit_posthocs library's posthoc_nemenyi_friedman function was used for pairwise comparison between models, with the confidence level set at 95%. The roc_ curve subroutine in scikit-learn1.2.1 library is used to generate ROC curve and calculate the area under the curve (AUC) to evaluate the efficiency of the optimal model [30]. The performance of the optimal model is evaluated by calling the pandas0.22.0 library output confusion matrix and its derived metrics. The measurement indexes included false positive (FP), false negative (FN), true positive (TP), true negative (TN), sensitivity (Sen), specificity (Spe), positive predictive value (PPV), negative predictive value (NPV), accuracy (ACC) and F1 Score [31].

# Result

Several AI models are generated by training ResNet18, MobileNet, DenseNet, AlexNet, and SqueezeNet. The average accuracy (ACC) and loss function values (LFV) obtained from 8-fold cross-validation (CV) are presented in Table 1. Significant statistical differences (P<0.01) were observed among the models generated by the five pre-trained networks for the four blood lipid indicators. Among these, the models generated by ResNet18, MobileNet, and DenseNet exhibited higher accuracy: [TG: ResNet18(LFV=0.101, ACC=0.97±0.012),MobileNet(LFV=0.115, ACC=0.976±0.011),DenseNet(LFV=0.111, ACC=0.975±0.011);TC: ResNet18(LFV=0.083, ACC=0.982±0.010),MobileNet(LFV=0.250, ACC=0.906±0.023),DenseNet(LFV=0.074, ACC=0.987±0.009);LDL-C: ResNet18(LFV=0.268, ACC=0.999±0.003),Mobile-Net(LFV=0.373, ACC=0.877±0.028),DenseNet(LFV=0.333, ACC=0.859±0.029);HDL-C: ResNet18(LFV=0.241, ACC=0.902±0.025),MobileNet(LFV=0.271, ACC=0.885±0.027),DenseNet(LFV=0.284, ACC=0.897±0.026)]. From Table 2, it can be seen that there are significant statistical differences between ResNet18 and MobileNet, AlexNet, SqueezeNet, but no statistical difference between ResNet18 and DenseNet in mean comparison. Thus, all the model input test sets of each network are tested, and the mean histograms of different models for predicting different blood lipid indexes (ACC, F1-score, PPV, TPR) are shown in Fig 4. It can be found that the performance of ResNet18 is the best. Therefore, we select the optimal model with the lowest loss function from each 8-fold CV model of ResNet18 as the final model, and output the confusion matrix and its derived metrics to evaluate the performance of the optimal model. The confusion matrix of the optimal model is shown in Fig 5, and the derived metrics are presented in Table 3:The classification accuracy (ACC) for TG was 0.853±0.025, with an F1-score of 0.885±0.023;For TC, ACC was 0.833±0.029, with an F1-score of 0.571±0.039;For LDL-C, ACC was 0.937±0.021, with an F1-score of 0.886±0.027;For HDL-C, ACC was 0.936±0.021, with an F1-score of 0.897±0.026.The corresponding ROC curve of the optimal model is shown in Fig 6.

# Discussion

## Clinical significance

Currently, the primary method for effective clinical monitoring is blood drawing. The advantage of this method is that it is low-cost and easy to perform; however, it requires invasive sampling of blood. In clinical diagnosis, in order to obtain

**Table 1. Average accuracy and loss function value of different pre-training Network models in K-fold Cross Verification.**

|  |  | Resnet18 | Mobilenet | Densenet | Alexnet | Squeezenet | Friedman test | P value |
|---|---|---|---|---|---|---|---|---|
| TG | LFV | 0.101 | 0.115 | 0.111 | 0.648 | 0.584 |  |  |
|  | ACC | 0.97±0.012 | 0.976±0.011 | 0.975±0.011 | 0.653±0.034 | 0.696±0.033 | 619.695 | <0.01 |
| TC | LFV | 0.083 | 0.250 | 0.074 | 0.390 | 0.251 |  |  |
|  | ACC | 0.982±0.010 | 0.906±0.023 | 0.987±0.009 | 0.873±0.026 | 0.919±0.021 | 546.415 | <0.01 |
| LDL-C | LFV | 0.268 | 0.373 | 0.333 | 0.438 | 0.420 |  |  |
|  | ACC | 0.999±0.003 | 0.877±0.028 | 0.859±0.029 | 0.861±0.029 | 0.868±0.028 | 309.510 | <0.01 |
| HDL-C | LFV | 0.241 | 0.271 | 0.284 | 0.610 | 0.692 |  |  |
|  | ACC | 0.902±0.025 | 0.885±0.027 | 0.897±0.026 | 0.623±0.041 | 0.576±0.042 | 439.818 | <0.01 |

**Table 2. P-values from Nemenyi test comparisons between Resnet18 and other pre-training Network models across different blood lipid indices.**

|  |  | Mobilenet | Densenet | Alexnet | Squeezenet |
|---|---|---|---|---|---|
| TG | Resnet18 | 0.023 | 0.7451 | <0.01 | <0.01 |
| TC | Resnet18 | <0.01 | 0.3991 | <0.01 | <0.01 |
| LDL-C | Resnet18 | <0.01 | <0.01 | <0.01 | <0.01 |
| HDL-C | Resnet18 | 0.1338 | 1.00 | <0.01 | <0.01 |

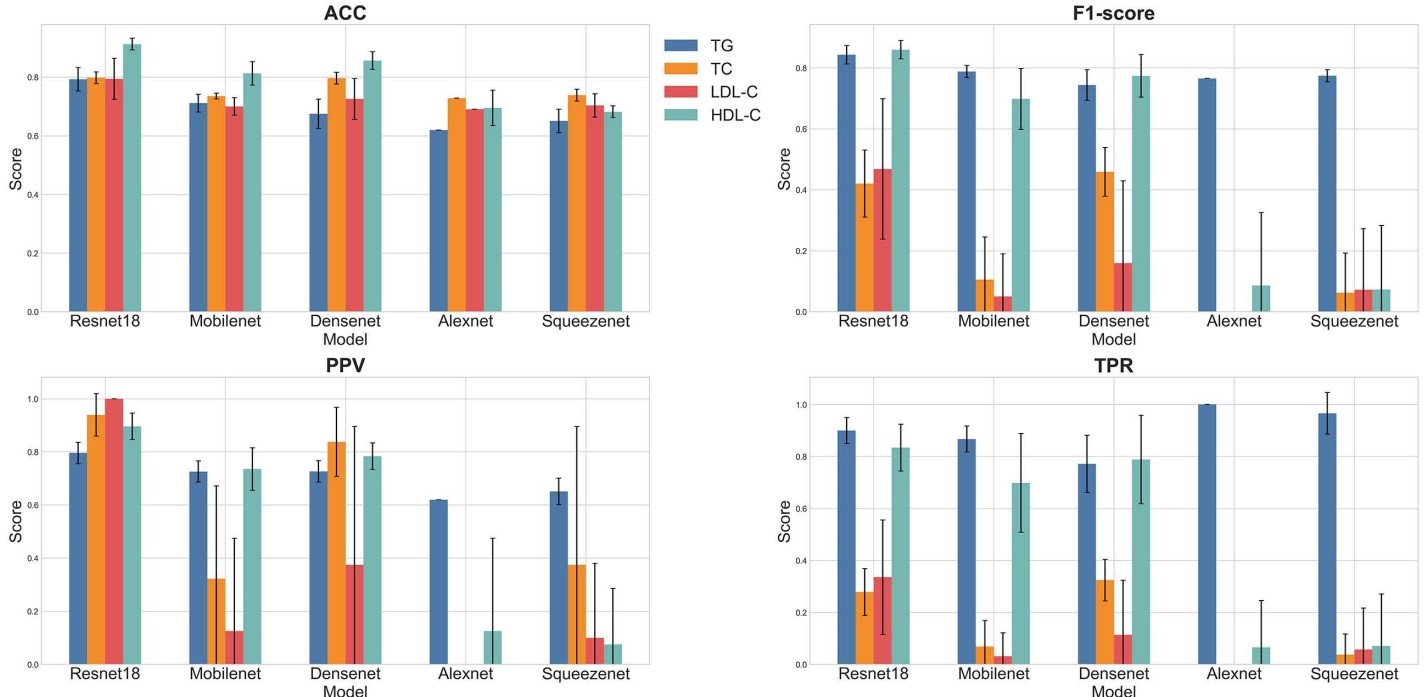

**Fig 4. Histogram of predictive efficiency of different models for different blood lipid indexes (ACC, F1-score, PPV, TPR).**

enough physiological and biochemical indicators, more and more sample tubes are used to draw blood. If imaging tests could predict blood lipid levels, it would reduce the need for multiple blood draws and provide early warnings for other diseases related to blood lipids.

Therefore, we aimed to use imaging examinations to predict the levels of routine clinical blood lipid items [10]. The fat maps generated from CSE-MRI examinations show significant potential in this endeavor due to their short scanning time and strong lipid signal representation [16,32–34]. Compared to ultrasound images, CSE-MRI fat maps are not influenced by cross-sectional position, and they are less affected by shimming compared to MRS, allowing for the generation of multiple images in a single session [35]. Combined with the progress of artificial intelligence in medical image classification and prediction, We trained multiple binary classification models using CSE-MRI fat maps. These models can be used to predict variations in TG, TC, LDL-C and HDL-C levels, thereby expanding the functionalities of CSE-MRI images and offering a novel method for obtaining blood lipid indices.Although, compared to standard blood tests, performing MRI solely for screening dyslipidemia is neither practical nor economical, it should be noted that our method is not proposed as a primary screening tool but rather as a way to add value to existing imaging data. Specifically, when patients undergo abdominal MRI for other reasons (e.g., assessing fatty liver, focal liver lesions, or other clinical indications) and CSE-MRI sequences are included, the acquired fat maps can be opportunistically analyzed by our AI models without any additional scanning time or cost, thereby providing incidental predictions of dyslipidemia risk as early warning signals and facilitating timely referrals for confirmatory blood tests. We believe this "opportunistic screening" model enhances the utility of routinely acquired clinical data. Our findings suggest this approach is promising and merits further investigation.

## Model design and tuning

Before model training, we classified the images using fasting blood levels of TG, TC, LDL-C, and HDL-C as the gold standard to simulate the most authentic clinical routine lipid measurement, thereby ensuring the accuracy of the samples. To

PLOS Digital Health

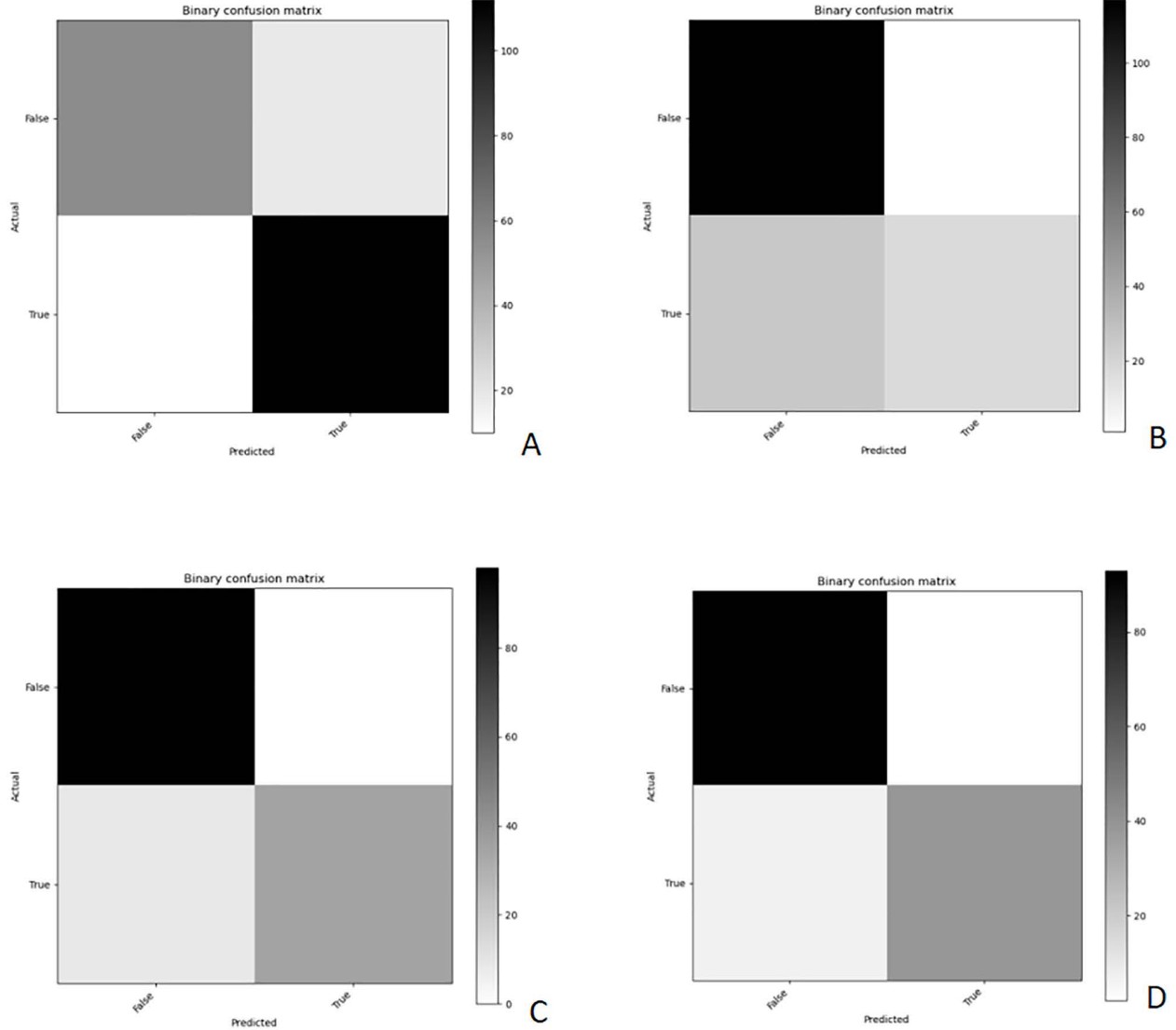

**Fig 5. Confusion Matrix Diagram generated by different Blood Lipid Index Test sets in Resnet18 optimal Model Test (A:TG, B:TC, C: LDL-C, D: HDL-C).**

**Table 3. Confusion Matrix and its Derived metrics generated by different Blood Lipid Index Test sets in Resnet18 optimal Model Test.**

|  | FN | FP | TN | TP | PPV | NPV | TNR | TPR | ACC | F1-score | Micro-AUC | Macro-AUC |
|---|---|---|---|---|---|---|---|---|---|---|---|---|
| TG | 10 | 19 | 56 | 112 | 0.855±0.025 | 0.848±0.026 | 0.747±0.031 | 0.918±0.020 | 0.853±0.025 | 0.885±0.023 | 0.890±0.022 | 0.870±0.024 |
| TC | 26 | 1 | 117 | 18 | 0.947±0.018 | 0.818±0.030 | 0.992±0.007 | 0.409±0.039 | 0.833±0.029 | 0.571±0.039 | 0.790±0.032 | 0.630±0.038 |
| LDL-C | 9 | 0 | 98 | 35 | 0.999±0.003 | 0.916±0.023 | 0.999±0.003 | 0.795±0.034 | 0.937±0.021 | 0.886±0.027 | 0.990±0.008 | 0.999±0.003 |
| HDL-C | 7 | 2 | 93 | 39 | 0.951±0.018 | 0.93±0.022 | 0.979±0.012 | 0.848±0.030 | 0.936±0.021 | 0.897±0.026 | 0.970±0.014 | 0.980±0.012 |

train AI models with the highest possible accuracy using small sample data, we undertook extensive preparation and optimization work. Firstly, we employed the k-fold cross-validation (CV) method for dataset allocation, which helps avoid overfitting due to insufficient validation data and allows for the evaluation of the model's generalization capability. Secondly, we

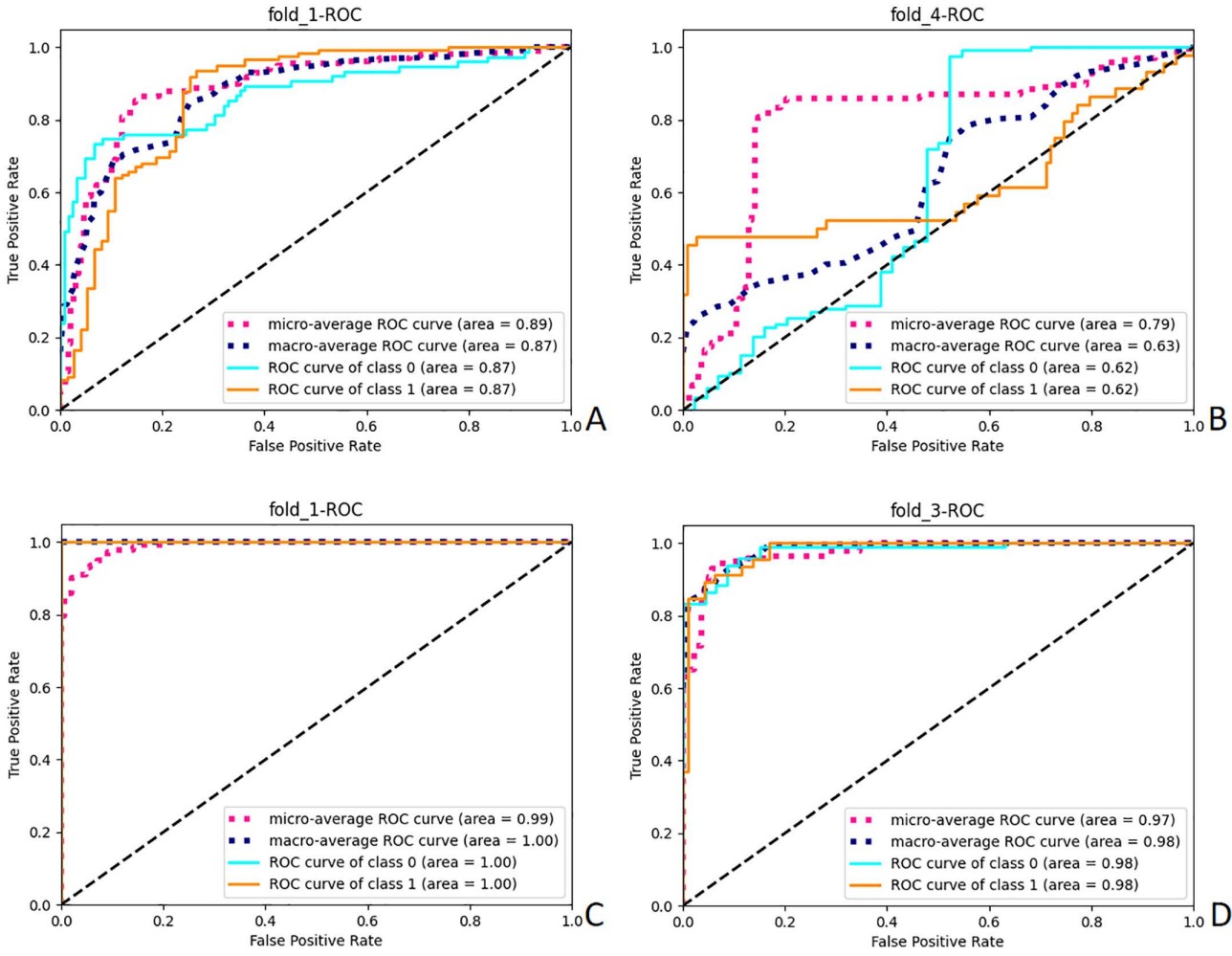

**Fig 6. ROC graphs generated by different Blood Lipid Index Test sets in Resnet18 optimal Model Test.** A: ROC curves generated by TG fold-1 model. B: ROC curves generated by TC fold-4 model. C:ROC curves generated by LDL-C fold-1 model. D:ROC curves generated by HDL-C fold-3 model.

compared numerous lightweight medical image classification models such as ResNet18, MobileNet, DenseNet, AlexNet, and SqueezeNet [23–25,36]. This selection was not arbitrary but was based on screening pre-trained networks that have demonstrated excellent performance in medical image classification tasks from numerous studies. Results indicated that ResNet18 achieved more outstanding outcomes, which the authors attribute to the small sample size and uneven distribution of positive rates in this study. Deep networks are prone to gradient vanishing during training, and ResNet is specifically designed to address this issue [36]. After reviewing previous studies [29,31], we chose the shallowest ResNet18 as the pre-trained network to further prevent overfitting. Additionally, during model training, we used L2 regularization to prevent the model from becoming excessively large or sparse [27], and dynamic testing based on the Adam adaptive learning rate optimizer [26] to avoid the uncertainty of manual parameter tuning. Furthermore, high-quality training equipment ensured that batch size and initial learning rate could be set very low. Notably, the optimal model generated after each iteration was saved, which, although resulting in a larger number of models and higher space occupancy, allows for the selection of the optimal model across different iteration counts. In summary, we utilized the best available technologies to enhance the model's accuracy, stability, and generalization capability.

## Model effectiveness analysis

In order to explore the applicability of different classification networks to the data set of this study, we compared many light-weight medical image classification models, and carried out K-fold cross-validation on each model. From the internal verification data in Table 1, we can see that the average performance of ResNet18 is similar to that of MobileNet and DenseNet in all indicators of TG, TC, LDL-C and HDL-C. All of them have higher accuracy and lower loss function, but these can not prove the degree of over-fitting of the model. Therefore, we further input the test set to test. According to the test results (Fig 4), ResNet18 balances accuracy and generalization ability, while showing a smaller error range, which may be related to its residual architecture that helps alleviate gradient vanishing. This is beneficial for training with a limited sample size. The average ACC in Fig 4A reveals significant performance differences among the five different pre-trained networks, with all models exhibiting some degree of overfitting. However, ResNet18's overfitting is significantly less than other models, as well illustrated by the F1-Score in Fig 4B. From the data distribution diagram of Fig 2, we can see that the data distribution of TC and LDL-C is to a great extent uneven, and the positive data of the training set is much smaller than the negative data, so the F1-Score is only 0.42 and 0.47. The TPR in Fig 4D also proves this. On the contrary, the numerical values of ACC, F1-Score, PPV and TPR of TG and HDL-C are very excellent, indicating that the prediction performance of TG and HDL-C of Resnet18 is relatively stable after migration learning.However, in terms of test set, it can be seen from Fig 5 and Table 3 that LDL-C achieves unexpected performance. Under the premise that the training data set is highly unevenly distributed, the model achieves further improved ACC (0.937) and F1 score (0.886), and the optimal model with TG and HDL-C also performs well. The PPV, TPR, AUC and other performance indexes of the above three optimal models are also higher, which eliminates the misleading accuracy of the model in the unbalanced data, but the performance of TC is still poor, and the F1-score only 0.571, indicating poor performance. Therefore, among the optimal classification models, the models for TG, LDL-C, and HDL-C based on ResNet18 already have certain practical value.

On the other hand, the classification efficiency of the TC models trained based on various pre-trained networks was very low. Although they achieved relatively high accuracy on the internal validation set, they exhibited almost no generalization capability in the test set. This issue is partly related to the uneven data distribution, but the clinical data we currently have is already quite challenging to obtain. More importantly, an analysis of the principles reveals that this is related to the various substances contained within total cholesterol (TC) itself. TC represents the sum of all cholesterol in the blood, including cholesterol in LDL-C, HDL-C, and VLDL-C [37]. Consequently, its characteristics are highly complex and influenced by the levels of LDL-C and HDL-C. Therefore, when the training sample size is small, the computer struggles to identify effective features that capture its variability.

## Future work and prospects

Overall, our work holds substantial practical value. Notably, the optimal model trained on TG, LDL-C, and HDL-C using ResNet18 demonstrated high classification performance. However, as indicated by the results, due to the limited amount of training data, the uneven distribution of data in some training sets, and the non-specific nature of the components that make up TC, the performance of the TC model was suboptimal. There is also room for further improvement in the prediction efficiency of TG, LDL-C and HDL-C. In future training processes, we should try to balance the uneven distribution of data sets, either through parameter optimization or by expanding both the single-class sample and the overall sample size, in order to train a better prediction model for clinical application.

## Conclusion

In this study, we designed and trained an artificial intelligence model using deep learning to predict abnormalities in blood lipid levels based on liver CSE-MRI fat maps. The performance of the optimal models for TG, LDL-C and HDL-C validate the effectiveness of our approach, which can be used as an early warning reference for clinical blood lipids. However, achieving accurate prediction for all blood lipid indices remains a significant challenge that requires further investigation.

# Author contributions

**Data curation:** Yina Wang, Xiao Xiao.

**Formal analysis:** Kai Deng.

**Methodology:** Xiaofan Chen.

**Resources:** Xiong Wu, Haitao Yang.

**Software:** Zhichao Feng, Jianmin Yuan.

**Supervision:** Bo Jiang, Weijun Situ, junjiao hu.

**Writing – original draft:** Bo Jiang.

**Writing – review & editing:** Xi Guo, junjiao hu.

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
