## [Decision Letter · Decision Letter 0]

10 Mar 2025

Response to Reviewers
Revised Manuscript with Track Changes
Manuscript
**Journal Requirements:**

1. Please provide a complete Data Availability Statement in the submission form, ensuring you include all necessary access information or a reason for why you are unable to make your data freely accessible. If your research concerns only data provided within your submission, please write "All data are in the manuscript and/or supporting information files" as your Data Availability Statement.

**Additional Editor Comments (if provided):**
**Reviewers' Comments:**

**Comments to the Author**

1. Does this manuscript meet PLOS Digital Health’s publication criteria?

Reviewer #1: Yes

Reviewer #2: Partly

2. Has the statistical analysis been performed appropriately and rigorously?

Reviewer #1: Yes

Reviewer #2: No

3. Have the authors made all data underlying the findings in their manuscript fully available (please refer to the Data Availability Statement at the start of the manuscript PDF file)?

Reviewer #1: Yes

Reviewer #2: Yes

4. Is the manuscript presented in an intelligible fashion and written in standard English?

Reviewer #1: Yes

Reviewer #2: Yes

Reviewer #1: Comments

1. Sample Size and Generalizability

• The study includes only 89 patients, which raises concerns about how well the findings can be generalized to a broader population.

• The class imbalance (e.g., significantly more cases with high TG than normal TG) could affect the model’s ability to perform consistently across different lipid levels.

• Suggestion: The authors should clarify whether they applied any techniques to address this imbalance (e.g., oversampling, weighted loss functions) and discuss its potential impact on model performance.

2. Model Selection and Justification

• The manuscript tests several AI models and selects ResNet18 as the best-performing option. However, the reasoning behind this choice is not fully explained.

• Suggestion: It would be helpful to discuss why ResNet18 performed better than other models and whether alternative architectures (e.g., newer deep learning methods) were considered.

3. Performance of the Model (Particularly for TC)

• The F1-score for total cholesterol (TC) is quite low (0.571), suggesting the model struggles to predict this parameter accurately.

• Suggestion: The authors should discuss potential reasons for this and whether modifications (e.g., feature engineering, different preprocessing techniques) could improve performance.

• Additionally, confidence intervals for AUC and accuracy values are missing, which are essential for assessing the robustness of the model.

4. Clinical Relevance and Validation

• While the model’s results are interesting, the lack of an external validation dataset is a significant limitation.

• Suggestion: The authors should acknowledge this and discuss the need for validation in a larger, independent cohort before considering clinical applications.

Reviewer #2: 1. The authors have responded with "ok" to the competing interests and data availability statements, which is insufficient. Please explicitly disclose any competing interests, if applicable, or state that none exist. Additionally, provide a clear and detailed data availability statement, specifying whether the data are publicly available, subject to restrictions, or not accessible, along with relevant repository links or access conditions.

2. Throughout the manuscript, the authors use `accuracy_score`, `f1_score`, `precision_score`, and `recall_score`. Please use the standard metric names (accuracy, F1 score, precision, recall) instead. In the abstract, simply refer to them as "accuracy metrics" rather than listing each one, as an exhaustive list is unnecessary.

3. Abstract Results: Mentioning transfer learning again in results is redundant. Perhaps, rephrase the first sentence. The authors could improve clarity and readability by restructuring the last sentence.

4. Introduction - Paragraph 1: Please provide a clear definition of dyslipidemia to enhance readability for a non-clinical audience.

5. Introduction - Paragraph 2: The last few lines lack proper citations. The authors should provide appropriate references to support the statements.

6. Introduction - Paragraph 3: The phrase "with the increasing demand for medical experience" seems a bit vague. It might be clearer to specify the increasing demand for "improved healthcare" or "more efficient healthcare services." "But it can not" should also be changed to "however, it cannot."

7. Introduction - Parahraph 3: The gap statement at the beginning is very vague, the authors should make it make it more objective and concrete.

8. Introduction - Paragraph 4: The first two lines are redundant.

9. Methods: The Experimental Design and Model Training sections provide overlapping information. It would be more efficient to remove the Experimental Design section and restructure the Methods section to begin with the Dataset Description, followed by a discussion of Model Development and Statistical Evaluation.

10. Methods - CSEMRI Scanning: In the imaging parameters, it would be helpful to include the number of slices for each MRI scan, as well as the slice thickness.

11. Methods: Repeatedly mentioning the libraries used throughout the manuscript is unnecessary, particularly for common ones like 'os' and 'glob.' It would be more efficient to make the code publicly available, which would allow readers to easily access the full list of libraries and dependencies.

12. Methods - Image Classification and Preprocessing: A total of 1,757 images were extracted from 89 MRIs, which were then split into training and test sets. It is unclear whether this split was random or done along patient lines (e.g., 19 MRIs in the test set and 70 in the training set, with images extracted accordingly). If the split was random, this could potentially lead to data leakage. The authors should explicitly clarify the method of data splitting. If the split was random, I recommend reseparating the data along patient lines and retraining the models to avoid potential bias.

13. Methods - Model Training: The authors mention '... and other lightweight medical image classification networks,' but the figures and tables only present the five architectures discussed in the manuscript. It would be helpful to clarify whether additional architectures were considered or to update the manuscript to reflect only the five mentioned.

14. Methods - Model Evaluation and Statistical Analysis: The details on how cross-validation works are unnecessary. This section lacks statistical rigor—was any statistical comparison performed between the models' performances to identify the best-performing model? Including such analysis would strengthen the manuscript.

15. Results: The Results section is lacking in detail and mainly repeats information from the Methods section. It needs to be rewritten to provide a more in-depth analysis and clear presentation of the findings.

16. Discussion: The entire Discussion section is highly redundant, with a substantial portion repeating information from the Methods section. The authors should focus on discussing the results in depth and emphasize the clinical applicability of their findings.

17. Figures 2 and 3 convey the same information. It would be more efficient to merge them into a single figure to avoid redundancy.

18. Figure 4 has several issues: there is a lot of overlapping values, the legend is repeated four times, the y-axis scale is inconsistent across plots, and the alignment is not optimal. Additionally, while the authors mention calculating the standard deviation, error bars are not presented in the figure.

19. The fonts in Figure 5 are too small, and there are no subplot titles.

20. No standard deviations are provided in any of the results tables. I also suggest that the authors include a comprehensive table in the supplementary materials with all metrics for the models' cross-validation results.

**Figure resubmission:****Reproducibility:** To enhance the reproducibility of your results, we recommend that authors of applicable studies deposit laboratory protocols in protocols.io, where a protocol can be assigned its own identifier (DOI) such that it can be cited independently in the future. Additionally, PLOS ONE offers an option to publish peer-reviewed clinical study protocols. Read more information on sharing protocols at https://plos.org/protocols?utm_medium=editorial-email&utm_source=authorletters&utm_campaign=protocols

---

## [Decision Letter · Decision Letter 1]

15 Aug 2025

Response to Reviewers
Revised Manuscript with Track Changes
Manuscript
**Journal Requirements:**
**Additional Editor Comments (if provided):**
**Reviewers' Comments:**

**Comments to the Author**

Reviewer #2: (No Response)

publication criteria?

Reviewer #2: Partly

3. Has the statistical analysis been performed appropriately and rigorously?

Reviewer #2: Yes

4. Have the authors made all data underlying the findings in their manuscript fully available (please refer to the Data Availability Statement at the start of the manuscript PDF file)?

Reviewer #2: Yes

5. Is the manuscript presented in an intelligible fashion and written in standard English?

Reviewer #2: No

Reviewer #2: 1. One of the major concern is the clinical relevance of this technology. It would be infeasible for patients to be subjected to expensive MR imaging solely to predict dyslipidemia risk. A definitive diagnosis of dyslipidemia would still be most effectively/efficiently done via blood draw since MR for something as common as dyslipidemia would be prohibitively expensive for most primary care providers to consider and would take away imaging time from acute patients. This would seem to be more of an incidental finding for when an abdominal MRI is clinically indicated for another complaint.

2. The Results section lacks sufficient quantitative and statistical detail. Multiple performance claims are made without reporting actual values (e.g., average accuracy, F1-score, PPV, TPR, AUC, etc.). While Tables 1 and 2 are referenced, key numerical findings should also be summarized in the text.

3. The narrative flow of the Results and Discussion sections is weak. The transition from model comparison to model selection and final evaluation is abrupt and lacks justification. The rationale for selecting ResNet18 over other models needs to be more comprehensive i.e., was this decision based solely on accuracy, or were other factors such as variance or training time considered? Additionally, Figure 3 should include appropriate error bars to support the comparison, and the narrative should be revised for clearer structure.

4. The choice of training for 40 epochs is not justified. Did the authors consider or implement early stopping to prevent overfitting?

5. Citations for biomarker cutoff values are missing. These should be clarified in the Methods section.

6. Figure formatting is inconsistent. Ensure uniform font size and style across all figures and verify that all axis labels and text elements are legible.

7. The term “Derivative Index” is unclear. The authors should define this term explicitly when it is first introduced in the Results section.

8. The manuscript contains numerous language and terminology issues. Ambiguous phrasing and redundant wording reduce clarity. Terminology should be corrected—for instance, "layer thickness" should be "slice thickness", "picture" should be "image", and "pieces" should be replaced with "instances", "images", or "patients" as appropriate. The manuscript also contains punctuation and grammatical errors that should be addressed through careful proofreading.

9. The second line in the Introduction section, seems unnecessary to the Introduction. Similarly, maternal disease is unclear, this could refer both to maternally inherited disorders or obstetric complications

10. In Introduction, second paragraph, the wording is confusing. The reader could interpret HDL as “reversing cholesterol being transported to the liver” which implies that HDL transports cholesterol to the peripheral tissues. However, HDL is generally associated with "cholesterol transport to" the liver.

**Do you want your identity to be public for this peer review?** For information about this choice, including consent withdrawal, please see our Privacy Policy

Reviewer #2: No

**Figure resubmission:****Reproducibility:** To enhance the reproducibility of your results, we recommend that authors of applicable studies deposit laboratory protocols in protocols.io, where a protocol can be assigned its own identifier (DOI) such that it can be cited independently in the future. Additionally, PLOS ONE offers an option to publish peer-reviewed clinical study protocols. Read more information on sharing protocols at https://plos.org/protocols?utm_medium=editorial-email&utm_source=authorletters&utm_campaign=protocols

---

## [Decision Letter · Decision Letter 2]

17 Nov 2025

Development and validation of an artificial intelligence model based on liver CSE-MRI fat maps for predicting dyslipidemia

PDIG-D-24-00452R2

Dear hujunjiao hu,

We are pleased to inform you that your manuscript 'Development and validation of an artificial intelligence model based on liver CSE-MRI fat maps for predicting dyslipidemia' has been provisionally accepted for publication in PLOS Digital Health.

Best regards,

Alexander Wong

Section Editor

PLOS Digital Health

**Additional Editor Comments (if provided):**

**Reviewer Comments (if any, and for reference):**

Reviewer's Responses to Questions

**Comments to the Author**

Reviewer #2: All comments have been addressed

publication criteria?

Reviewer #2: Partly

3. Has the statistical analysis been performed appropriately and rigorously?

Reviewer #2: Yes

4. Have the authors made all data underlying the findings in their manuscript fully available (please refer to the Data Availability Statement at the start of the manuscript PDF file)?

Reviewer #2: Yes

5. Is the manuscript presented in an intelligible fashion and written in standard English?

Reviewer #2: Yes

Reviewer #2: Please add a properly formatted legend to Figure 4 with an appropriate font size.

**Do you want your identity to be public for this peer review?** For information about this choice, including consent withdrawal, please see our Privacy Policy

Reviewer #2: No
